# MC-JEPA: A Joint-Embedding Predictive Architecture for Self-Supervised Learning of Motion and Content Features

## Abstract

Self-supervised learning of visual representations has been focusing on learning content features, which do not capture object motion or location, and focus on identifying and differentiating objects in images and videos. On the other hand, optical flow estimation is a task that does not involve understanding the content of the images on which it is estimated. We unify the two approaches and introduce MC-JEPA, a joint-embedding predictive architecture and self-supervised learning approach to jointly learn optical flow and content features within a shared encoder, demonstrating that the two associated objectives; the optical flow estimation objective and the self-supervised learning objective; benefit from each other and thus learn content features that incorporate motion information. The proposed approach achieves performance on-par with existing unsupervised optical flow benchmarks, as well as with common self-supervised learning approaches on downstream tasks such as semantic segmentation of images and videos.

## 1 Introduction

Self-supervised learning in vision has been dominated lately by approaches that aim at learning content features; i.e. features containing information that allows to identify and differentiate objects; in images (Chen et al., 2020a; Grill et al., 2020; Chen & He, 2020; Zbontar et al., 2021; Bardes et al., 2022a; Caron et al., 2020; 2021; Zhou et al., 2022; Assran et al., 2022; 2023), or videos (Qian et al., 2021; Recasens et al., 2021; Feichtenhofer et al., 2021; Tong et al., 2022). Most methods focus on learning global features that achieve strong results in tasks such as object classification or action recognition in videos. A more recent trend aims at learning localized features, that perform well on local tasks such as detection and segmentation (Xiao et al., 2021; Wang et al., 2021; Hénaff et al., 2021; 2022; Bardes et al., 2022b). However, these methods focus on understanding the content of images and videos and are not able to learn information at the pixel level, such as motion in videos or details in textures. In this paper, we focus on jointly learning motion features by using self-supervised optical flow estimation (Horn & Schunck., 1981) from videos as a pretext task, and content features with general self-supervised learning.

The Optical flow captures the motion, or dense-pixel correspondence, that occurs between two images, for instance consecutive frames in a video, or images from a stereo pair. Estimating it is a fundamental problem in computer vision, whose solution is key to tasks such as visual odometry, depth estimation, or object tracking. Classical approaches cast optical flow estimation as an optimization problem (Horn & Schunck., 1981; Brox et al., 2004), where the objective is to match pixels with a smoothness constraint. Approaches based on neural networks and supervised learning (Yu et al., 2016; Ilg et al., 2017; Hui et al., 2018; Sun et al., 2018; Yang & Ramanan, 2019; Zhao et al., 2020; Teed & Deng, 2020; Jiang et al., 2021; Bai et al., 2022), are limited by the difficulty of labelling data in the real world, compared to using synthetic data. Self-supervised methods allow learning from large collections of real-world video data (Ren et al., 2017; Liu et al., 2019b;a; Zhong et al., 2019; Im et al., 2020; Liu et al., 2020; Luo et al., 2021; Jonschkowski et al., 2020; Stone et al., 2021) and offer an alternative that is now competitive with supervised approaches. However, most current methods only focus on motion without relying on the (semantic) content of the video, a problem that we solve by learning motion and content features in images at the same time with a multi-task approach.

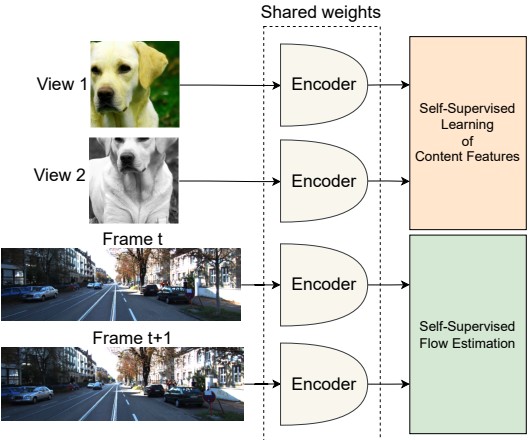

Figure 1: **Multi-task self-supervised learning of content and motion features.** MC-JEPA combines a self-supervised features learning and optical flow estimation approach in a multi-task setup where with a single shared encoder. The self-supervised learning of content features objective is trained on ImageNet and the self-supervised flow estimation task is trained on various videos datasets. Our final encoder produces features that have motion and content information, and that can be used to estimate optical flow in videos or for content understanding downstream tasks.

Recent techniques learn spatial correspondences between video frames (Jabri et al., 2020; Bian et al., 2022; Xu & Wang, 2021; Tokmakov et al., 2022). The goal is to track the location of objects and therefore capture content information that optical flow estimation does not. These approaches can be seen as object-level motion estimation. They learn features that are very specific to the tracking task, with very poor generalization to other visual downstream tasks. Very often, they are trained on small video datasets that are not as diverse as large image datasets such as ImageNet (Deng et al., 2009), which reinforces the poor quality of the visual features learned. A more reliable way to build visual representations is to learn multiple tasks at the same time (Zhang et al., 2021; Girdhar et al., 2022). We thus propose MC-JEPA (Motion-Content Joint-Embedding Predictive Architecture), a method that learns optical flow estimation and content features, in a multi-task setting with a shared encoder, with a joint-embedding-predictive architecture (LeCun, 2022). Our contributions can be summarized as follows:

- We propose a method for learning self-supervised optical flow from synthetic and real video data, based on PWC-Net (Sun et al., 2018), and improved with several additional components such as a backward consistency loss and a variance-covariance regularization term. We call this first method M-JEPA.
- We combine M-JEPA in a multi-task setup with VICReg (Bardes et al., 2022a), a self-supervised learning method trained on ImageNet, in order to improve our estimated flow, and produce content features that transfer well on many downstream tasks. Our final method is called MC-JEPA.
- We evaluated MC-JEPA on a range of optical flow benchmarks such as KITTI 2015 (Menze & Geiger, 2015) and Sintel (Butler et al., 2012), image and video segmentation tasks on Cityscapes (Cordts et al., 2016) or DAVIS (Pont-Tuset et al., 2017), and demonstrate strong performance on all these tasks with a single encoder.

We hope that MC-JEPA will be a first step towards self-supervised learning approaches that are based on multi-task learning and joint-embedding architectures, and that can be trained on any visual data, images or video, and that generalizes well on a wide range of tasks, from motion prediction tasks to content understanding tasks.

## 2 RELATED WORK

**Self-supervised learning.** The recent advances in self-supervised learning have been mainly driven by the general approach of learning invariances to hand-crafted data augmentations, using a joint-

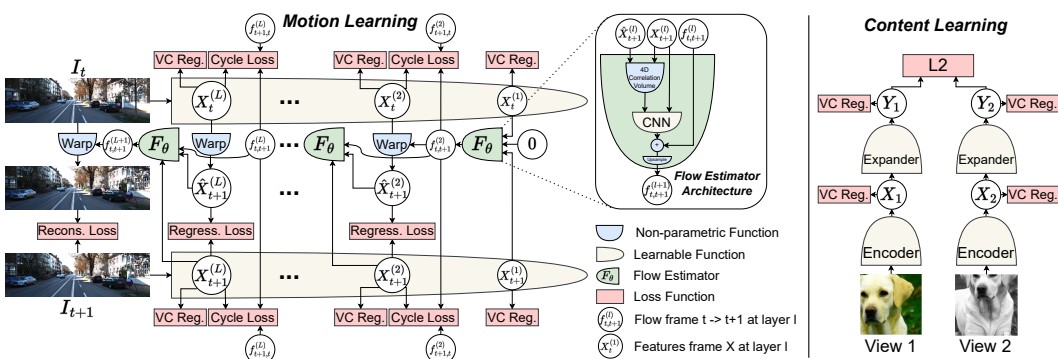

Figure 2: **MC-JEPA architecture.** Our method learns motion through optical flow estimation on videos and content through joint-embedding of views of images, in a multi-task way with a shared encoder. Our optical flow estimation architecture is based on PWC-Net (Sun et al., 2018) and works as follows. Given a pair of consecutive frames $I_t$, $I_{t+1}$ in a video, an encoder produces a set of pyramidal features $\{X_t^{(l)}\}$ and $\{X_{t+1}^{(l)}\}$. The flow is estimated in a coarse-to-fine manner, starting at the lowest resolution features $X^{(1)}$. A first flow $f_{t,t+1}^2$ is estimated by the flow estimator network, then used to warp the features $X_t^{(2)}$, which is compared to $X_{t+1}^{(2)}$ with a regression loss. The flow is then iteratively refined at every layer by predicting the residual flow and adding it to the previous layer flow. The final flow is used to warp $I_t$ and compare the warped image with $I_{t+1}$ using a reconstruction loss. Forward-backward flow consistency is encouraged with the cycle consistency losses, which minimizes the distance between $X_t^{(l)}$ and $f_{t,t+1}^{(l)}(f_{t+1,t}^{(l)}(X_t^{(l)}))$ at every layer. When the encoder is trained in the multi-task setup with a standard self-supervised learning criterion, the training is very unstable, which is prevented by the variance-covariance regularization term on every feature layer.

embedding architecture (LeCun, 2022). Among self-supervised learning methods learning from images, contrastive methods push together concepts that are visually close and push away concepts that are different in the embedding space (Hjelm et al., 2019; Chen et al., 2020a; He et al., 2020; Chen et al., 2020b; Mitrovic et al., 2021; Dwibedi et al., 2021; Chen et al., 2021; Tomasev et al., 2022; Li et al., 2022), clustering methods categorized embeddings into a balanced set of clusters (Caron et al., 2018; 2020; 2021), non-contrastive methods either prevent collapsing solutions with architectural tricks (Grill et al., 2020; Lee et al., 2021; Chen & He, 2020), or with covariance-based regularization (Ermolov et al., 2021; Zbontar et al., 2021; Bardes et al., 2022a; Garrido et al., 2023b), which is equivalent under some assumptions to contrastive methods (Garrido et al., 2023a). Finally, some methods are based on masking and patch-reconstruction (Bao et al., 2022; He et al., 2022; Zhou et al., 2022; Assran et al., 2022; 2023). These methods focus on learning a global representation of the input, which is best suited for classification tasks. Dense self-supervised learning rather focuses on learning local features (Xie et al., 2021; Wang et al., 2021; Xiao et al., 2021; Yang et al., 2021; Wang et al., 2022; Yang et al., 2022; Hénaff et al., 2021; 2022; Chen et al., 2022; Caron et al., 2023), which is best suited for detection and segmentation downstream tasks. The loss functions and methods developed with images have led to the application of similar approaches to videos (Qian et al., 2021; Recasens et al., 2021; Feichtenhofer et al., 2021; Tong et al., 2022; Parthasarathy et al., 2022), with the objective of learning a representation that transfers well on action recognition benchmarks.

**Optical flow estimation.** Classical techniques for optical flow estimation are based on the optimization of a matching term and a smoothness term for a given pair of images, without any kind of learning (Horn & Schunck., 1981; Brox et al., 2004; Sun et al., 2010). Later, methods based on supervised learning and convolutional neural networks came, first without any prior knowledge in architecture (Yu et al., 2016; Ilg et al., 2017), then specifically designed to tackle flow estimation (Ranjan & Black, 2017; Sun et al., 2018; Yang & Ramanan, 2019; Teed & Deng, 2020). Supervised flow estimation is limited to learning with synthetic data, and unsupervised flow estimation is a promising direction towards learning on any video data. Photometric consistency was introduced by (Ren et al., 2017) and is at the basis of every unsupervised optical flow estimation method.

Additional self-supervision signals can be found with distillation of reliable matches (Liu et al., 2019b;a), global geometric constraint (Zhong et al., 2019), or data augmentation consistency (Liu et al., 2020; Stone et al., 2021). Fusing multi-layer similarities (Im et al., 2020) and carefully designing the interpolation for upsampling (Luo et al., 2021) further improve the estimated flow quality. Finally, a comprehensive set of additional tricks that help unsupervised optical flow is presented in (Jonschkowski et al., 2020).

**Learning correspondences.** Learning from videos has been focusing on learning a global representation for a video, but another interesting task is learning spatial correspondences between consecutive frames. A promising direction for learning these correspondences is contrastive random walks (Jabri et al., 2020), which can also be done at the pixel level (Bian et al., 2022). Correspondences can also be learned at the object level (Xu & Wang, 2021; Patrick et al., 2021), or combined with a memory (Tokmakov et al., 2022), in order to deal with occluded objects. Learning optical flow can be seen as learning correspondences at the pixel-level, which is not captured by popular self-supervised learning methods.

**Multi-task Learning.** Multi-task learning is commonly used to train an encoder on multiple tasks, when the different tasks benefit from each other. Several works use it to learn a shared representation between images and videos (Zhang et al., 2021; Girdhar et al., 2022). However, very few works use multi-task learning for self-supervised learning, the idea was introduced in (Doersch & Zisserman, 2017) and used for anomaly detection tasks in (Georgescu et al., 2021), without many follow-up work. We simply use multi-task learning for learning self-supervised content features and optical flow at the same time with a single shared encoder.

## 3    PROPOSED APPROACH

In this section we describe our architecture and improvements for self-supervised optical flow estimation with a hierarchical coarse-to-fine approach, the loss functions of our method, our self-supervised general objective and multi-task setup, our data sampling strategy, and a set of tricks for stabilizing training. Section 3.1 introduces our M-JEPA method for optical flow estimation, and Section 3.2 presents how we combine M-JEPA with multi-task learning into our final MC-JEPA method.

### 3.1    OPTICAL FLOW

Given a pair of RGB images, $I_t, I_{t+1} \in \mathbb{R}^{3,H,W}$, the corresponding optical flow is defined by the correspondence map $f \in \mathbb{R}^{2,H,W}$, that for a given position in $I_t$, denotes the position of the corresponding pixel in $I_{t+1}$. The goal is to learn a flow estimator function $F_\theta$ with parameters $\theta$, which outputs the flow for a pair of images $f = F_\theta(I_t, I_{t+1})$, by training it on a set of image sequences $D = \{\{I_t\}_{t=1}^T\}_{i=1}^N$. Unsupervised flow estimation usually works with a regression loss, or photometric consistency loss, which ensures that the image $I_t$ warped by the predicted flow $f$ is consistent with $I_{t+1}$, and a regularizer that encourages $f$ to be smooth. Most methods differ in the way these terms are implemented, in the details of the encoder and flow estimator architecture, and in additional self-supervisory signals.

**Regression and smoothness.** We use the coarse-to-fine hierarchical flow estimator PWC-Net (Sun et al., 2018), which we adapt to work with our custom encoder architecture described in Appendix C. Given a set of features $X_t^{(l)}, X_{t+1}^{(l)} \in \mathbb{R}^{d^{(l)} \times h^{(l)} \times w^{(l)}}$, corresponding to level $l$ of pyramids for images $I_t$ and $I_{t+1}$ with $l \in \{1, ..., L\}$, we first estimate a flow $f_{t,t+1}^{(2)} = F_\theta(X_t^{(1)}, X_{t+1}^{(1)}, 0)$, then recursively refine this flow at higher and higher resolutions by predicting the residual flow at every layer:

$$f_{t,t+1}^{(l+1)} = F_\theta(X_t^{(l)}, X_{t+1}^{(l)}, f_{t,t+1}^{(l)}). \tag{1}$$

Our estimator $F_\theta(X_t, X_{t+1}, f)$ works as follows. First the feature $X_t$ is warped as $\hat{X}_{t+1} = f(X_t)$, then a 4D correlation volume $V = \hat{X}_{t+1} X_{t+1}^T$ is calculated and is fed to a small convolutional network $g_\phi(V, X_t, \hat{X}_{t+1}, f)$ which predicts the residual flow. We then use a multi-scale loss on the intermediate feature layers of the encoder, defined as follows:

$$\mathcal{L}_{\text{reg}} = \sum_{l=1}^L \|X_{t+1}^{(l)} - \hat{X}_{t+1}^{(l)}\|_2^2, \tag{2}$$

Table 1: **Quantitative results.** Comparison of the performance of our model on: (1) Sintel (Butler et al., 2012) clean and final, and KITTI 2015 (Menze & Geiger, 2015) optical flow estimation benchmarks; (2) Pascal VOC (Everingham et al., 2010), Cityscapes (Cordts et al., 2016) and ADE20k (Zhou et al., 2019), both frozen and fine-tune linear segmentation benchmarks; (3) DAVIS-2017 (Pont-Tuset et al., 2017) and video object segmentation benchmark, against several self-supervised methods optimized for a single task specifically. EPE is the average end-point-error ($\downarrow$ Lower is better). F1 is the average-f1 error in (%) ($\uparrow$ Lower is better). mIoU is the mean intersection-over-union ($\uparrow$ Higher is better). $(\mathcal{J}\&\mathcal{F})_m$ is the average between mean region similarity and mean contour-based accuracy ($\uparrow$ Higher is better). MC-JEPA is our full model trained in multi-task way on ImageNet and flow estimation. M-JEPA is our model without content learning, trained only on flow estimation. The best and second best result for each benchmark are **bold** and underlined.

| | | Optical Flow Estimation | | | | | | Image Segmentation | | | | | | Video Seg. |
| | | Sintel Clean | | Sintel Final | | KITTI 2015 | | Pascal VOC | | CityScapes | | ADE20k | | Davis 2017 |
| Method | Backbone | train EPE | test EPE | train EPE | test EPE | train EPE | test F1 | Frozen mIoU | FT mIoU | Frozen mIoU | FT mIoU | Frozen mIoU | FT mIoU | $(\mathcal{J}\&\mathcal{F})_m$ |
|---|---|---|---|---|---|---|---|---|---|---|---|---|---|---|
| Rand. weights | CNX-T | 23.71 | - | 24.02 | - | 24.88 | - | 0.5 | - | - | - | - | - | - |
| *flow methods* | | | | | | | | | | | | | | |
| UFlow (Jonschkowski et al., 2020) | PWC | 2.50 | 5.21 | 3.39 | 6.50 | 2.71 | 11.13 | 7.8 | - | - | - | - | - | 42.0 |
| ARFLow (Liu et al., 2020) | PWC | 2.79 | 4.78 | 3.73 | 5.89 | 2.85 | 11.80 | 7.9 | - | - | - | - | - | - |
| UPFlow (Luo et al., 2021) | PWC | 2.33 | 4.68 | 2.67 | 5.32 | 2.45 | 9.38 | 8.8 | - | - | - | - | - | - |
| SMURF (Stone et al., 2021) | RAFT | 1.71 | 3.15 | 2.58 | 4.18 | 2.00 | 6.83 | 10.4 | - | - | - | - | - | - |
| *correspondence methods* | | | | | | | | | | | | | | |
| VFS (Xu & Wang, 2021) | R50 | - | - | - | - | - | - | 51.2 | - | - | - | - | - | 68.9 |
| MCRW (Bian et al., 2022) | PWC | 2.84 | 5.68 | 3.82 | 6.72 | 2.81 | 11.67 | 39.8 | - | - | - | - | - | 57.9 |
| *content methods* | | | | | | | | | | | | | | |
| VICReg (Bardes et al., 2022a) | CNX-T | - | - | - | - | 13.5 | - | 60.1 | 77.8 | 59.8 | 76.3 | 28.6 | 41.1 | 58.1 |
| VICRegL (Bardes et al., 2022b) | CNX-T | - | - | - | - | 11.4 | - | 66.8 | 79.7 | 64.9 | 78.3 | 30.6 | 44.1 | 66.7 |
| MoCo v3 (Chen et al., 2021) | ViT-S | - | - | - | - | 12.9 | - | 57.1 | 75.9 | 56.5 | 74.0 | 23.7 | 39.8 | - |
| DINO (Caron et al., 2021) | ViT-S | - | - | - | - | 11.8 | - | 65.2 | 79.5 | 64.8 | 78.1 | 30.5 | 43.5 | 69.9 |
| *ours* | | | | | | | | | | | | | | |
| M-JEPA | CNX-T | 2.98 | - | 3.82 | - | 3.01 | - | 9.4 | - | - | - | - | - | - |
| MC-JEPA | CNX-T | 2.81 | 5.01 | 3.51 | 6.12 | 2.67 | 11.33 | 67.1 | 79.9 | 65.5 | 78.4 | 30.8 | 44.2 | 70.5 |

and a reconstruction loss on the last layer that is at the image level:

$$\mathcal{L}_{\mathrm{rec}} = d(I_{t+1}, \hat{I}_{t+1}), \tag{3}$$

where $d$ is a loss function that is a linear combination of an $l2$, $l1$, and SSIM losses. In addition, we use the smoothness regularizer of (Jonschkowski et al., 2020) that constrains the produced flow to be smooth, and allows us to deal with repetitive or textureless paterns:

$$\mathcal{L}_{\mathrm{smooth}} = \sum_{d \in \{x,y\}} \sum_{p} \exp(-\lambda \nabla_d I) \|\nabla_d f_{t,t+1}\|_1, \tag{4}$$

where x and y are directions in which the predicted flow is constrained to remain stable if the image gradient does not significantly change.

**Cycle consistency.** Flow estimation is a non-symmetric operation, as not all pixels of $I_t$ have a correspondence in $I_{t+1}$ and vice versa. For a given pair of images, we estimate both the forward and backward flows. We introduce a cycle-consistency loss that constraint the features $X_t$ warped by $f_{t,t+1}$ then by $f_{t+1,t}$ to match with $X_t$, the loss is defined as follows:

$$\mathcal{L}_{\mathrm{cycle}} = \sum_{l=1}^{L} \|X_t^{(l)} - f_{t+1,t}(f_{t,t+1}(X_t^{(l)}))\|_2^2, \tag{5}$$

where $f(X)$ is the warping operation of $X$ by flow $f$. We symmetrize the loss and do the same for $X_{t+1}$. In order to deal with occlusion, we follow (Liu et al., 2019a) and use forward-backward compatibility, only applying $\mathcal{L}_{\mathrm{reg}}$ on the pixels that have a correspondence in both the forward and the backward flows.

**Variance-covariance regularization.** Finally, in order to regularize the features produced by our encoder, we introduce a variance-covariance regularization loss function (Bardes et al., 2022a), de-

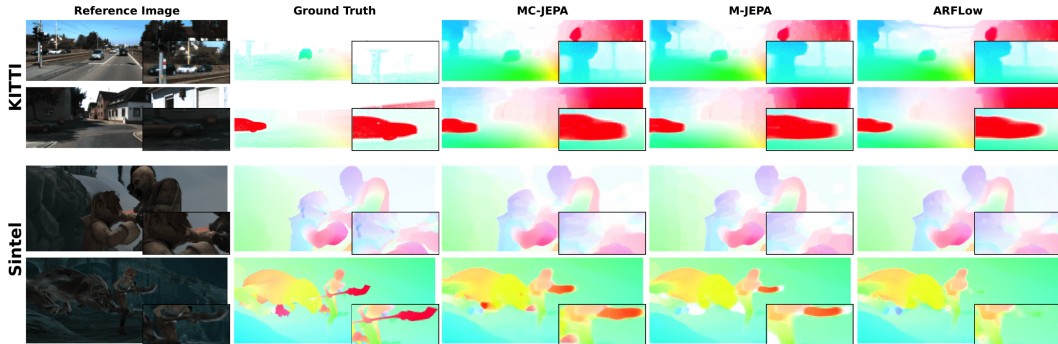

Figure 3: **Qualitative visualization: optical flow.** We compare our results of our complete model (MC-JEPA) and our model only pretrained on flow (M-JEPA) with ARFlow. Top 2 rows are from KITTI-15, bottom 2 rows are from Sintel clean and Sintel final.

fined as follows:

$$
\begin{aligned}
\mathcal{L}_{\text{vc}} = \sum_{l=1}^{L} \frac{1}{d} \sum_{j=1}^{d} \max(0, \gamma - \sqrt{\text{Var}(X_{t,j}^{(l)}) + \epsilon}) \\
+ \frac{1}{d} \sum_{i \neq j} [C(X_t^{(l)})]_{i,j}^2 .
\end{aligned}
\tag{6}
$$

where $\text{Var}$ is the empirical variance and $C$ is the empirical covariance matrix after centering the features. This loss helps stabilizing the training with the multi-task setup described in Section 3.2, and also improves the performance of the method as shown by Table 11.

## 3.2 MULTI-TASK SELF-SUPERVISED LEARNING

This section describes how we combine M-JEPA with content learning into our final MC-JEPA method.

**Learning content features.** We follow the literature (Chen et al., 2020a; Grill et al., 2020; Caron et al., 2020; Bardes et al., 2022a) and learn content features by simply pre-training our encoder to jointly-embed two views of an image. We generate the views using image transformation such as random cropping and color jittering. In particular, we use the VICReg objective (Bardes et al., 2022a) and follow its protocol. From a seed image sampled in an unlabelled training dataset $\mathcal{D}$, two views are generated using common data augmentation such as random croppring and color jittering, the views are then rescaled to a fixed size and fed to an encoder, then mapped to an expander network on which the VICReg loss is applied. The VICReg loss $\mathcal{L}_{\text{ssl}}$ is similar to Eq. (6), with in addition an invariance term ($l_2$ loss) that makes the embedding of the two views closer to each other and is minimized over $\mathcal{D}$.

**Multi-task learning.** At a given iteration of training, we sample a batch of sequences from our video dataset and compute the flow loss, then sample a batch of images from ImageNet and compute our self-supervised learning loss, and then add the two losses and back-propagate the gradients into our encoder, expander, and flow estimator network. The encoder architecture and weights are shared between the two tasks. We illustrate our approach in Figure 1 for the general idea and Figure 2 for the detailed architecture. The final loss function that MC-JEPA optimizes is as defined follows:

$$
\sum_{\mathcal{D}_1} \mathcal{L}_{\text{rec}} + \mathcal{L}_{\text{reg}} + \mathcal{L}_{\text{smooth}} + \mathcal{L}_{\text{cycle}} + \mathcal{L}_{\text{vc}} + \sum_{\mathcal{D}_2} \mathcal{L}_{\text{ssl}},
\tag{7}
$$

where $\mathcal{D}_1$ is our video sequences dataset and $\mathcal{D}_2$ is our image dataset. The losses are balanced with additional coefficients that we tune carefully. Additional details are given in Appendix B, including the values we use for these coefficients.

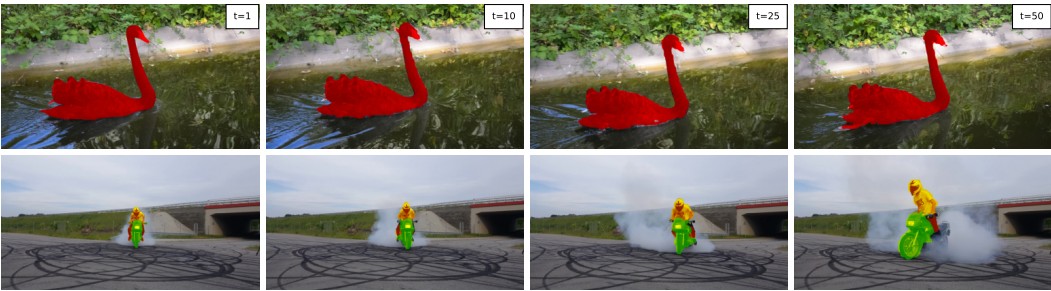

Figure 4: **Qualitative visualization: video segmentation.** We visualize the segmentation maps obtained by the frozen features learnt with MC-JEPA on the video instance tracking task on DAVIS 2017, for several video sequences, at frames t=1,10,25,50. Frame 1 is given as ground truth, and the others are predicted by our model.

## 4 EXPERIMENTS

### 4.1 DATASETS

Our model is pretrained in a single phase on a set of datasets commonly used for optical flow estimation, as well as on ImageNet-1k (Deng et al., 2009). Our video and flow datasets are KITTI (raw (A. et al., 2013), 2012 multiview (Geiger et al., 2012) and 2015 multiview (Menze & Geiger, 2015)), MPI Sintel (Butler et al., 2012) (clean, final and raw movie), FlyingChairs (Yu et al., 2016), FlyingThings (N. et al., 2016), and HD1K (D. et al., 2016). We evaluate the quality of our estimated flow on Sintel clean and final and KITTI 2015 and compare our model with state-of-the-art methods in self-supervised flow estimation. We evaluate the quality of our features on instance segmentation on Pascal VOC (Everingham et al., 2010), CityScapes (Cordts et al., 2016) and ADE20k (Zhou et al., 2019), both in linear frozen and fine-tuning evaluation. Finally, we evaluate our model on the DAVIS 2017 (Pont-Tuset et al., 2017) video segmentation and instance tracking benchmark popularized by (Caron et al., 2021).

### 4.2 MAIN RESULTS

**Optical flow.** We compare the flow estimated by our model with several state-of-the-art methods optimized for flow estimation, as well as with MCRW, which discovers the flow by learning contrastive random walks between pixels. Table 1 presents our results, which are on par with UFLow (Jonschkowski et al., 2020), ARFlow (Liu et al., 2020) and UPFLow (Luo et al., 2021), which are all optimized for flow estimation. SMURF (Stone et al., 2021) is better on all the benchmarks, but our goal is not to learn the best flow possible but rather to use it as a pretext task to learning general features and motion. However, we outperform MCRW which shares the same goal. Figure 3 presents our optical flow qualitative results.

**Instance Segmentation.** Table 1 presents the performance of MC-JEPA in various frozen and fine-tuned linear segmentation tasks, which are commonly used to evaluate the quality of the features learned by self-supervised learning models (Zhou et al., 2022; Bardes et al., 2022b). We outperform MoCo v3 (Chen et al., 2021) and VICReg (Bardes et al., 2022a), which is the method we use for our content features learning, by a large margin, which indicates that our flow estimation pretext task significantly helps the localization. Our results are on-par with VICRegL (Bardes et al., 2022b) which is specialized for segmentation and DINO (Caron et al., 2021) which has among the best self-supervised features available.

**Video Segmentation.** Finally, we compare the performance of MC-JEPA on a video segmentation instance tracking task on the DAVIS 2017 dataset, against VFS (Xu & Wang, 2021) and MCRW (Bian et al., 2022) which are correspondence learning methods and DINO. We outperform all these methods, which shows that learning motion through flow estimation is a good way of improving the learning of content features for tasks that requires motion information. Figure 4 shows qualitative results on DAVIS 2017. Overall, our method allows us to train a single model that performs very well on all the above-mentioned tasks, whereas all the concurrent works are specialized for either content feature learning or motion and optical flow estimation learning.

Table 2: **Ablation: flow datasets.** Impact on performance when varying the set of pretraining datasets. KITTI means pretraining on KITTI raw, 2012 and 2015. Sintel means pretraining Sintel raw, clean and final. FT/FC are FlyingThings and FlyingChairs. The metric for K15 (KITTI 2015), clean and final is the EPE. ISeg is the linear frozen evaluation on Pascal VOC, in mIoU, VSeg is the evaluation on DAVIS 2017, in $(\mathcal{J}\&\mathcal{F})_m$.

| KITTI | Sintel | FT/FC | HD1k | K15 | clean | final | ISeg | VSeg |
|---|---|---|---|---|---|---|---|---|
| ✓ | | | | 2.93 | 3.23 | 3.96 | 66.8 | 70.0 |
| | ✓ | | | 3.78 | 2.95 | 3.61 | 66.4 | 69.9 |
| ✓ | ✓ | | | 2.91 | 2.99 | 3.70 | 67.2 | 70.4 |
| ✓ | ✓ | ✓ | | 2.88 | 2.93 | 3.66 | 67.1 | 70.3 |
| ✓ | ✓ | ✓ | ✓ | 2.67 | 2.81 | 3.51 | 67.1 | 70.5 |

Table 3: **Ablation: estimator architecture.** Comparison between different flow estimator size form of normalization. The factor size influences the number of filters in each convolution of the estimator. LN means layer norm means usage of layer norm after every layer of the estimator, except the last one. l2 means l2-normalization before the last layer of the estimator.

| Factor size | #Params | LN | l2 | K15 | clean | final | ISeg | VSeg |
|---|---|---|---|---|---|---|---|---|
| 1 | 2M | | | | | crashed | | |
| 1 | 2M | ✓ | | 2.68 | 2.88 | 3.57 | 67.0 | 70.2 |
| 1 | 2M | | ✓ | 6.21 | 6.04 | 6.99 | 53.2 | 47.9 |
| 1 | 2M | ✓ | ✓ | 4.55 | 4.47 | 5.66 | 62.3 | 63.6 |
| 2 | 8M | ✓ | | 2.67 | 2.81 | 3.51 | 67.1 | 70.5 |

## 4.3 ABLATIONS

We perform many ablations on the components and training procedure of MC-JEPA , and evaluate our models on KITTI 2015 train (K15 in tables, metric is EPE), Sintel clean and final (clean and final in tables, metric is EPE), Pascal VOC linear frozen evaluation (ISeg in tables, metric is mIoU), and DAVIS 2017 video segmentation (VSeg in tables, metric is $(\mathcal{J}\&\mathcal{F})_m$, which are all relatively fast to perform.

**Flow datasets.** We start by evaluating the effect of varying the set of data used flow estimation. Table 2 presents our results when incorporating or not various datasets. As expected, training on only KITTI or Sintel offers great performance in their respective evaluation set. Progressively adding FlyingChairs and Things, and HD1k, improves the flow results, but has very little influence on the segmentation tasks. The benefit on segmentation from doing flow estimation is independent from the domain on which the flow estimator is trained.

**Flow estimator architecture.** When pretraining in our multi-task setup with ImageNet we observed many instabilities related to the gradient and the exploding norm of the estimator, and that we describe in Section A. We tried several changes to the flow estimator architecture to overcome these issues, namely using LayerNorm and l2-normalization. Table 3 presents our results when incorporating these elements, as well as when increasing the size of the estimator. Not regularizing the estimator led to crashing runs. $l2$-normalization is very inefficient, as it constrains the last layer to directly produce flows in the correct range of values. Using LayerNorm is the best solution and effectively prevents the estimator from exploding norms and gradients. Increasing the size of the estimator marginally improves the results.

**Backbone.** Our backbone is a ConvNeXt-T (Liu et al., 2022), we study the impact of pretraining models with other backbones, in particular ResNet-50, and the backbone of PWC-Net (Sun et al., 2018) commonly used by concurrent flow estimation methods. Table 4 presents our results. The original PWC backbone is not adapted to learn good content features, and Resnet-50 results are not as good as ConvNeXt-T results.

**Data sampling.** We experiment with different strategies for sampling the data. For a simple baseline, we use a pretrained self-supervised model in ImageNet and train the flow estimator on top of the frozen features, or by fine-tuning the model. We demonstrate the usefulness of multi-task learning by playing with various other strategies; either we alternate between one epoch of ImageNet learning and one epoch of flow estimation, or we alternate between one batch of each, or we finally sample a batch from each, and back-propagate through the addition of the losses. Table 5 presents our results for each strategy. Training the flow estimator on top of frozen features is too hard of a constraint, but even when fine-tuning is done, optimizing the flow estimation task degrades the performance on segmentation too much. Alternating between epochs is not optimal, and the best solution is to alternate between batches and even combine the losses for optimal flow estimation results.

Table 4: **Ablation: backbone.** Comparison of the performance of MC-JEPA when using different backbones.

| Backbone | #Params | K15 | clean | final | ISeg | VSeg |
|---|---|---|---|---|---|---|
| PWC-Net | 8M | 2.66 | 2.80 | 3.47 | 14.8 | 10.1 |
| ResNet-50 | 21M | 2.71 | 2.85 | 3.59 | 55.8 | 60.1 |
| ConvNeXt-T | 23M | 2.67 | 2.81 | 3.51 | 67.1 | 70.5 |

Table 5: **Ablation: data sampling.** Comparison between different training order and data sampling strategies.

| Strategy | K15 | clean | final | ISeg | VSeg |
|---|---|---|---|---|---|
| Flow estimator training | 13.52 | 13.82 | 14.81 | 60.1 | 65.2 |
| Flow estimator fine-tuning | 2.71 | 2.82 | 3.77 | 61.3 | 62.3 |
| Epoch alternation | 4.54 | 4.91 | 5.57 | 63.5 | 66.9 |
| Batch alternation | 2.78 | 2.95 | 3.62 | 67.1 | 70.5 |
| Combined loss | 2.67 | 2.81 | 3.51 | 67.1 | 70.5 |

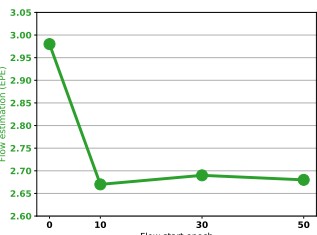 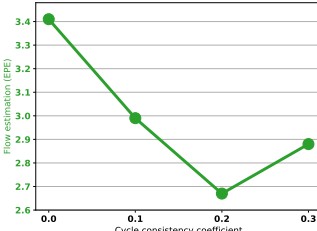 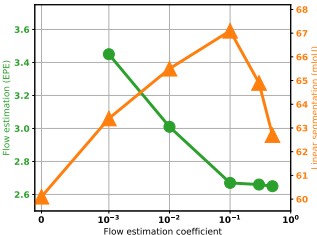

Figure 5: (1) **Ablation: flow start epoch**. Flow estimation performance as a function of the ImageNet training epoch from which flow estimation starts. There are 100 pretraining epochs in total. (2) **Ablation: cycle consistency coefficient.** Flow estimation performance as a function of the coefficient used to balance the cycle consistency loss of Eq (5). (3) **Ablation: multi-task balancing coefficient.** Flow estimation and segmentation performance as a function of the balancing coefficient between flow losses and SSL loss in Eq (7).

**Flow start epoch.** We found that starting multi-task learning of flow and content features at the beginning of training was not necessary, as the features are changing very fast, and we only start with ImageNet pretraining and introduce flow estimation after a given number of epochs. Figure 5 (1) shows that starting after 10 epochs of ImageNet pretraining is the best among several values, when the total number of epochs is fixed to 100. Starting later and doing fewer flow estimation epochs saves a lot of computation time while giving similar results.

**Cycle consistency.** Figure 5 (2) shows an ablation on the cycle consistency coefficient that controls the importance of the cycle consistency loss of Eq (5). Introducing the loss significantly improves the flow estimation, which is explained by the fact that it adds an additional constraint on the embeddings to be predictable from each other. The coefficient needs to be carefully tuned, as the performance is very sensitive to it.

**Multi-task balancing coefficient.** Figure 5 (3) shows an ablation on the multi-task coefficient that balances our flow estimation loss and our content features loss. We already observe a significant improvement when introducing flow estimation, even with a very small coefficient. As we increase the coefficient, both the flow estimation and segmentation improve until we reach a threshold (0.1), after which the segmentation results degrade a lot. This shows that even if flow estimation improves the segmentation performance, there is a trade-off between learning motion and content features, and tuning the multi-task coefficient is crucial to maintain a strong level of performance for both.

## 5 CONCLUSION

We have introduced MC-JEPA, a multi-task approach to learning of motion and content features with self-supervised learning and optical flow estimation. MC-JEPA performs well in a wide variety of tasks, ranging from optical flow estimation to segmentation of images and videos. We hope that our approach will foster the use of multi-task learning in self-supervised learning, which might be a path towards learning features that generalize to any downstream task. Future work will learn motion and content from larger collections of natural videos and train the two objectives in a shared data domain, capturing short- and long-range interactions in a hierarchical way.

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
