## A  IMPLEMENTATION DETAILS

**Data sampling strategy.** Most optical flow estimation models are trained in a curriculum-learning way by starting on the hardest synthetic datasets, or on a large collection of data, and are then fine-tune iteratively on each target domain data. For instance, they are pretrained on the Sintel raw Butler et al. (2012), KITTI raw A. et al. (2013), or chairs dataset Yu et al. (2016), and are then specifically fine-tuned on the KITTI multi-view extension Menze & Geiger (2015) or the Sintel clean and final dataset. We do everything at the same time, we start by doing only SSL training on ImageNet Deng et al. (2009), and during pretraining, at a given epoch which is an hyper-parameter of our model, we introduce flow training with a mixture of all our training datasets at the same time. We randomly sample one batch of sequences from one of our video datasets at each iteration. Doing so allows us to remain general and not design the training procedure specifically for a given set of video datasets, nor changing the training recipe when adding or removing a dataset from the collection of pretraining datasets.

**Training stability.** Stabilizing the training is particularly challenging, as the two tasks are difficult to optimize together. The norm and gradient of the flow estimator weights explode rapidly, causing 'NaN' values that propagate to the loss. There are four essential parts of our method that help to completely remove these instabilities. We introduce LayerNorm Ba et al. (2016) layers in the flow estimator network, clip the value of the flow output to a valid flow range and carefully tune the weight decay and the learning rate of the flow estimator network. Additional details about training are given in Appendix B, and about the architecture in Appendix C.

**Encoder details.** We modify the ConvNeXt Liu et al. (2022) architecture, in particular the ConvNeXt-T model with 21M parameters, and adapt it to produce a set of pyramidal features with six levels, with a resolution that doubles between each level. We replace the stem layer with kernel size 4 of ConvNeXt by two convolutional layers with kernels of size 2. We describe our modifications precisely in Appendix C.

**Training details.** We train our model on 8 Nvidia Tesla V100-32Gb GPUs, with the AdamW optimizer Loshchilov & Hutter (2019), a weight decay of $1e-6$, a batch size of 384 and a learning rate of $3e-4$ for the encoder and $1e-4$ for the flow estimator. The learning rate follows a cosine decay schedule, starting from 0 with 10 warmup epochs and with final value of $1e-5$. The flow estimation objective is trained after 10 epochs of only pretraining on ImageNet. The expander architecture follows Bardes et al. (2022a) and is a fully-connected network with dimensions (768-8192-8192-8192). We give a complete description of our training hyper-parameters in Appendix B and the architecture of the flow estimator in Appendix C.

## B  HYPER-PARAMETERS

We provide in Table 6 the set of all hyper-parameters used to trained our MC-JEPA and M-JEPA models. We follow the data augmentation protocol of Bardes et al. (2022a) and generate 2 views by random cropping with uniform distribution cropping size, and random color jittering. We use the same image resolution as Teed & Deng (2020) for the flow datasets. We found that having specific optimization hyper-parameters for the flow estimator network and the backbone was beneficial for the performance and stability of the training. We therefore first tune the hyper-parameters with M-JEPA on flow estimation only and then fix the flow optimization hyper-parameters and tune the general self-supervised learning optimization hyper-parameters of MC-JEPA. We increase the flow consistency factor, which gave us a better performance, and decrease the clipping value of the outputed flow from 256 to 128, which was necessary for the stability of the training. We detail in Table 7 the precise list of all the datasets that we use to train the flow estimation objective. The final dataset is built by repeating each of the datasets a given number of time that is indicated in the Table. During training of MC-JEPA, we shuffle this dataset and sample one batch from it for every batch that is sampled from ImageNet. Finally, Table 8 details the coefficients that are used for the variance and covariance losses of Eq. (6), invariance loss, and flow loss of Eq. (7), at each layer of the architecture. As we go further into the network we increase the coefficients, because we found that the last layers need more regularization than the early layer. We do the the tuning in a greedy way, cross-validating each layer coefficient one by one.

Table 6: **Hyper-parameters.** List of all the hyper-parameters used for MC-JEPA and M-JEPA training. The values that are noted "-" indicate that the corresponding parameter is not used.

| Hyper-parameter | MC-JEPA | M-JEPA |
|---|---|---|
| *data* | | |
| ImageNet res. | (224, 224) | - |
| min_scale_crops | 0.08 | - |
| max_scale_crops | 1.0 | - |
| Sintel res. | (384, 832) | (384, 832) |
| KITTI res. | (256, 832) | (256, 832) |
| FlyingX res. | (384, 512) | (384, 512) |
| *optimization* | | |
| num_gpus | 8 | 8 |
| epochs | 100 | 100 |
| warmump_epochs | 10 | 10 |
| batch_size | 384 | - |
| optimizer | AdamW | AdamW |
| lr | $3e-4$ | - |
| scheduler | cosine | cosine |
| end_lr | $3e-8$ | - |
| weight_decay | $1e-6$ | - |
| betas | (0.9, 0.999) | (0.9, 0.999) |
| *architecture* | | |
| drop_path_rate | 0.1 | 0.1 |
| layer_scale_init_value | 0.0 | 0.0 |
| expander dims. | 8192-8192-8192 | - |
| *flow* | | |
| flow_alpha | 0.1 | 1.0 |
| flow_coeff | 1.0 | 1.0 |
| flow_clip_value | 128.0 | 256.0 |
| flow_start_epoch | 10 | 0 |
| flow_loss_smooth_factor | 75.0 | 75.0 |
| flow_cycle_consistency_coeff | 0.2 | 0.1 |
| flow_batch_size | 8 | 8 |
| flow_lr | $1e-4$ | $1e-4$ |
| flow_weigth_decay | $1e-6$ | $1e-6$ |

Table 7: **Flow datasets.** List of all the datasets that we use for training the flow estimation objective. The size is the total number of pairs on which the optical flow is estimated. Repetition is the number of time the dataset is repeated to build the final dataset from which a random batch is sampled at each iteration.

| Dataset | Size | Repetition |
|---------|------|------------|
| FlyingThings | 40302 | 1 |
| FlyingChairs | 22232 | 1 |
| KITTI raw | 42382 | 1 |
| KITTI 2012 train | 200 | 100 |
| KITTI 2012 multiview train | 3800 | 5 |
| KITTI 2012 val | 198 | 100 |
| KITTI 2012 multiview val | 3762 | 5 |
| KITTI 2015 train | 200 | 100 |
| KITTI 2015 multiview train | 3800 | 5 |
| KITTI 2015 val | 198 | 100 |
| KITTI 2015 multiview val | 3762 | 5 |
| Sintel raw. | 27858 | 1 |
| Sintel clean | 1041 | 5 |
| Sintel final | 1041 | 5 |
| HD1k | 1047 | 5 |

Table 8: **Loss coefficients.** The coefficients for each loss at each specific layer of the architecture. Var and Cov are the variance and covariance regularization losses of Eq. (6). Invaraince is the invariance loss of VICReg used only for self-supervised training on ImageNet. Flow is the sum of all the flow losses, without the SSL loss, in Eq. (7). L1 to L6 are the features stages from lowest to highest resolution of our modified ConvNeXt-T backbone.

| Layer | Var | Cov | Invariance | Flow |
|-------|-----|-----|------------|------|
| Expander output | 25.0 | 1.0 | 1.0 | - |
| Encoder output (L1) | 0.01 | 0.04 | - | 1.0 |
| L2 | 0.01 | 0.04 | - | 1.0 |
| L3 | 0.01 | 0.001 | - | 1.0 |
| L4 | 0.01 | 0.0 | - | 1.0 |
| L5 | 0.001 | 0.0 | - | 0.1 |
| L6 | 0.0001 | 0.0 | - | 0.01 |

## C  ARCHITECTURE DETAILS

We describe in Figure 6 the modifications we make from the ConvNeXt-T architecture. We modify the stem layer, with the objective of increasing from 5 to 6 the number of layers in our pyramidal features, and to have the final flow prediction layer be right after the first layer of the architecture, which makes it close to the pixels space. We split the stem convolutional layer with a wide kernel of $7 * 7$ into two smaller layers with kernel sizes $3 * 3$ and $4 * 4$, and reduce the stride value from 4, to 2 for each layer, in order to make the flow regression process smoother from one step to the next. We describe in Figure 7 the modifications we make to the PWC flow estimator architecture, we add a LayerNorm after each convolutional layer except the last one, which greatly improves training stability, and multiply the number of filters of these layers by a factor $C$ that we fix to 2 in practice for our final model. Not having a LayerNorm after the final layer of the flow estimator is essential as it would bias the flow values toward a range that is different from the possible flow values range.

## D  ADDITIONAL RESULTS

We provide in Table 9 additional metrics, in Figure 8 additional visualizations, on the optical flow benchmarks; and in Table 10 additional metrics, in Figure 9 additional visualizations, on the video segmentation task on DAVIS.

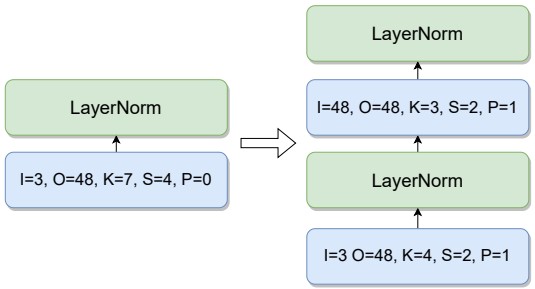

Figure 6: **Backbone stem modifications.** The stem convolutional layer in the ConvNeXt-T architecture is replaced in our backbone with two convolutional layers by reducing the kernel size and stride. Blue boxes are convolutional layers. I: input channels, O: output channels, K: kernel_size, S: stride, P: padding.

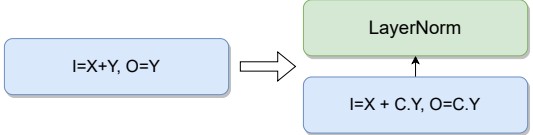

Figure 7: **PWC estimator modifications.** A layer norm layer is added after each convolutional layer of the PWC estimator architecture, the coefficient $C$ corresponds to the size factor parameter discussed in Table 3. We use $C = 2$ in practice. Blue boxes are convolutional layers. I: input channels, O: output channels.

Table 9: **Additional optical flow results.** We report additionally to Table 1, EPE performance on non-occluded (noc) and occluded (occ) pixels.

| | KITTI 2015 | | | Sintel Clean | | | Sintel Final | | |
|---|---|---|---|---|---|---|---|---|---|
| Method | all | noc | occ | all | noc | occ | all | noc | occ |
| M-JPEA | 3.01 | 2.26 | 6.98 | 2.98 | 1.54 | 23.99 | 3.82 | 2.17 | 24.68 |
| MC-JEPA | 2.67 | 2.08 | 6.24 | 2.81 | 1.25 | 23.82 | 3.51 | 1.99 | 24.23 |

Table 10: **Additional video segmentation results.** We report additionally to Table 1, mean region similarity $\mathcal{J}_m$ and mean contour-based accuracy $\mathcal{F}_m$.

| Method | $(\mathcal{J}\&\mathcal{F})_m$ | $\mathcal{J}_m$ | $\mathcal{F}_m$ |
|---|---|---|---|
| VICReg | 58.1 | 56.4 | 59.8 |
| VICRegL | 66.7 | 64.5 | 68.9 |
| DINO | 69.9 | 66.6 | 73.1 |
| MC-JEPA | 70.5 | 67.0 | 74.0 |

## E  ADDITIONAL ABLATION

One of the main issues with self-supervised learning is the collapse problem, where the network outputs a trivial solution. We prevent collapse by using variance-covariance (VC) regularization Bardes et al. (2022a) and apply it at every layer of our architecture to deal with stability issues when working in the multi-task setup. Table 11 presents an ablation of whether to use VC or not, and whether to use it only at the last layer, which is enough to prevent collapse, or at every layer of the architecture, and gives the best results for both flow estimation and segmentation. We also experiment with VC warming, which consists of training the VC layers during a given number of epochs before starting regular training, which helps fix stability issues and accelerate the convergence speed. Our results show that doing a single epoch of warmup is enough and helps the performance.

Table 11: **Ablation: variance-covariance.** Influence of variance-covariance regularization (VC) on performance. During warmup epochs, only the variance-covariance criteria are trained; this helps stabilizing the training. VC is applied in the last layer, in the expander output, or in every layer, with carefully chosen coefficients as described in Appendix B.

| Warmup ep. | Setup | K15 | clean | final | ISeg | VSeg |
|---|---|---|---|---|---|---|
| - | None | 3.41 | 3.37 | 4.45 | 47.3 | 37.8 |
| 0 | Last layer | 2.77 | 2.88 | 3.55 | 65.6 | 69.2 |
| 0 | All layers | 2.65 | 2.80 | 3.48 | 66.2 | 69.4 |
| 1 | All layers | 2.67 | 2.81 | 3.51 | 67.1 | 70.5 |
| 2 | All layers | 2.91 | 2.99 | 3.78 | 62.5 | 64.1 |

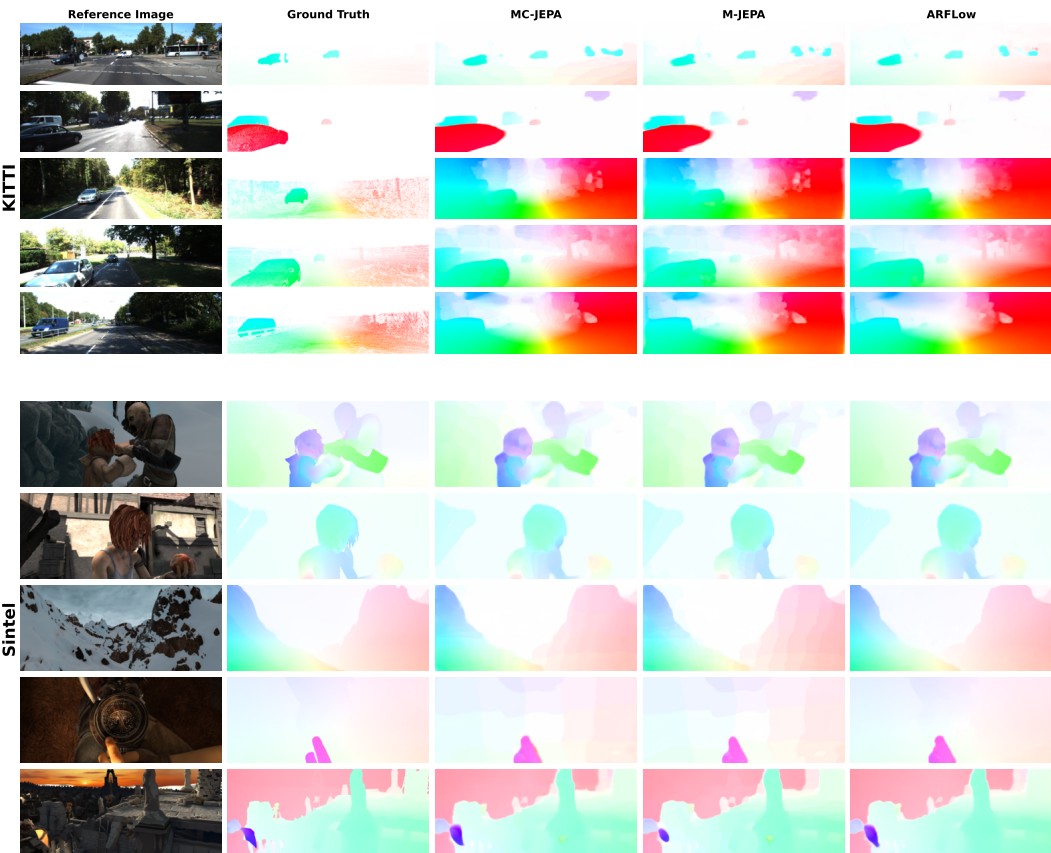

Figure 8: **Qualitative visualization: optical flow.** We compare our results of our complete model (MC-JEPA) and our model only pretrained on flow (M-JEPA) with ARFlow. Top 5 rows are from KITTI-15, bottom 5 rows are from Sintel.

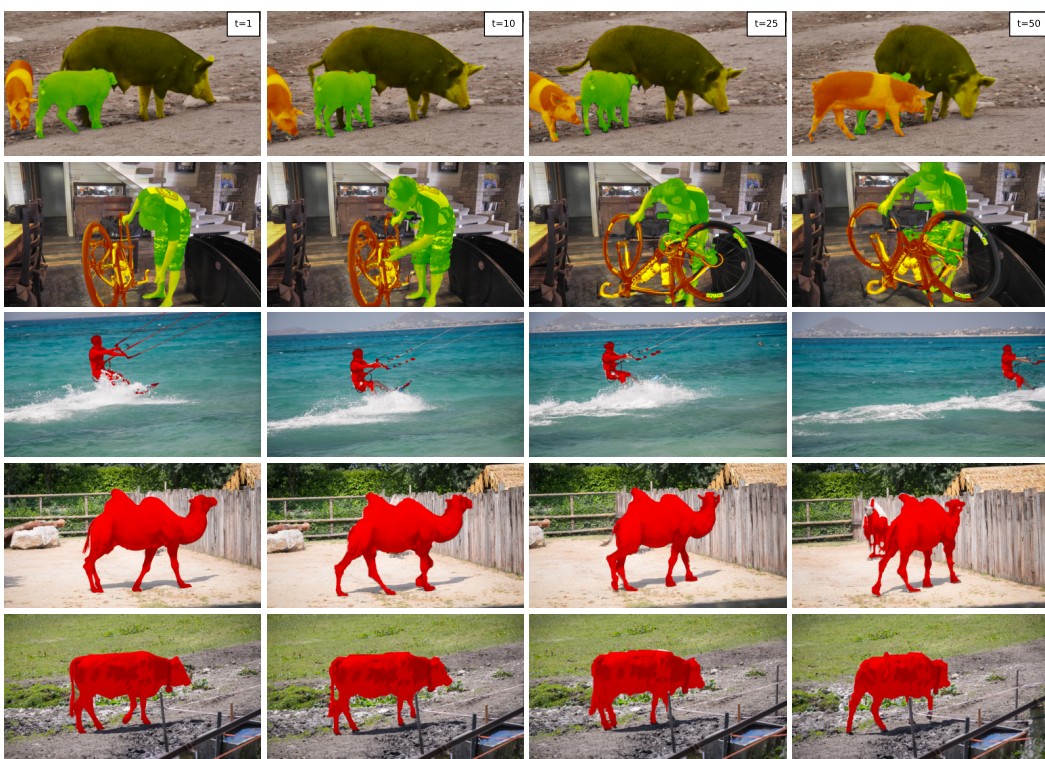

Figure 9: **Qualitative visualization: video segmentation.** We visualize the segmentation maps obtained by the frozen features learnt with MC-JEPA on the video instance tracking task on DAVIS 2017, for several video sequences, at frames t=1,10,25,50. Frame 1 is given as ground truth, and the others are predicted by our model.