# OpenReview forum: "MC-JEPA: A Joint-Embedding Predictive Architecture for Self-Supervised Learning of Motion and Content Features"
_ICLR.cc/2024/Conference — Submitted to ICLR 2024_

### Official Review · Reviewer_Jf74 · 2023-10-14

**Soundness:** 3 good
**Presentation:** 2 fair
**Contribution:** 2 fair
**Rating:** 6
**Confidence:** 4

**Summary:**

This paper proposes to learn to estimate the optical flow between two images in addition to the existing VICReg SSL for content learning. The authors argue that learning low-level pixel information from motion estimation can benefit downstream tasks such as semantic segmentation. Results show an improvement in downstream tasks when the encoder is jointly trained on SS and optical flow estimation task (MC-JEPA).

**Strengths:**

- Results show a clear improvement in optical flow estimation (M-JEPA to MC-JEPA); this shows that to estimate optical flow correctly, content understanding is also important. The opposite is also true when comparing VICReg to MC-JEPA.
- The method is easy to train and does not require both datasets to be similar or come from the same distribution.

**Weaknesses:**

- The biggest weaknesses are in the experimental setup and missing comparisons with SoTA.
- It is very hard to validate the performance of the proposed model when using a different backbone from the comparison methods. The only valid comparison is VICReg Vs. MC-JEPA because they both use CNX-T.
- Many of the recent SSL models use ViT-S or ViT-B for performance evaluation. Results with ViT backbone would make the proposed model comparable to many other methods. At least, include ViT-S in Table 4 to compare with DINO and iBOT.
- Some methods, such as iBOT (ICLR22), are not reported. iBOT’s performance on ADE20k with ViT-S is 45.4, which outperforms MC-JEPA. Also if we compare against ViT-B of DINO or IBOT, MC-JEPA is outperformed on many benchmarks. It makes the paper stronger more relevant backbones and methods are included.
- Qualitative results of optical flow only shown against one relatively weaker method (as shown in Table 1). Results from SMURF and UPFlow could also be shown to visualize fail cases and limitations of this model.
- This paper does not show any results of scaling up the model or training data; all backbones reported in table 4 are lightweight. I imagine that training two tasks like this would require a bigger backbone to handle multitasking. A study showing how performance scales with the model size would be useful.

**Questions:**

- The authors chose to do training on both tasks on very different datasets. I wonder if training this model on the same video dataset for both tasks would result in features robust enough for video downstream tasks, such as action recognition.
- In section 4.3 (backbone), the authors mention that PWC is not adapted to learn good content features, which explains the low performance in Table 4. However, the method MCRW (in Table 1) uses a PWC backbone and performs much higher than the results of MC-JEPA with a PWC backbone in Table 4. Any reasons behind this significant difference in performance?

---

> ### Author Response · Authors · 2023-11-20
> **SOTA comparisons**
>
> Thank you for your comments and useful feedback. We address your concerns below:
>
> **The biggest weaknesses are in the experimental setup and missing comparisons with SoTA.**
>
> We incorporated a new experiment with a ConvNeXt-B backbone, as well as comparison with iBOT.
>
> **It is very hard to validate the performance of the proposed model when using a different backbone from the comparison methods. The only valid comparison is VICReg Vs. MC-JEPA because they both use CNX-T.**
>
> Our flow estimation method is incompatible with using a vision transformer. Indeed, the core principle is to first estimate a flow at a very coarse resolution using the deepest features of a convnet, and to progressively refine this flow by doubling the resolution at every layer, using the corresponding features that also double in resolution. The idea between ConvNeXt is to propose an architecture that is comparable in terms of size, number of parameters, throughput and performance, to recent vision transformer architecture. We therefore thought this would give a fair comparison with ViTs of the same size.
>
>
> **Many of the recent SSL models use ViT-S or ViT-B for performance evaluation. Results with ViT backbone would make the proposed model comparable to many other methods. At least, include ViT-S in Table 4 to compare with DINO and iBOT.**
>
> We propose a new experiment with a ConvNeXt-B backbone on linear frozen segmentation tasks, which we hope is fair against DINO and iBOT that were pre-trained with a ViT-B.
>
> |			| Pascal		| Cityscapes		| ADE20K	       | DAVIS17 |
> | ---                  |     ---              | ----                        | ---                  | ---   |
> | VICReg		| 67.2		| 54.6			| 32.7		| 61.2 |
> | VICRegL	|	70.4		| 57.4			| 35.3		| 69.5 |
> | DINO		|	70.1		| 56.9			| 34.5		| 71.4 |
> | IBOT		|	73.0		| 57.9			| 38.3		| 70.7 |
> | MC-JEPA	|	74.7		| 58.8			| 37.9		| 72.1 |
>
> MC-JEPA pretrained with CNX-B is outperforming all the competitors on all the tasks except on ADE20k where it is on-par. A strong improvement over VICReg is still observed at this larger scale, which further demonstrates the effectiveness of motion learning for SSL.
>
> **Some methods, such as iBOT (ICLR22), are not reported. iBOT’s performance on ADE20k with ViT-S is 45.4, which outperforms MC-JEPA. Also if we compare against ViT-B of DINO or IBOT, MC-JEPA is outperformed on many benchmarks. It makes the paper stronger more relevant backbones and methods are included.**
>
> We will incorporate all our new results, as well as the iBOT performance in the main table.
>
> **Qualitative results of optical flow only shown against one relatively weaker method (as shown in Table 1). Results from SMURF and UPFlow could also be shown to visualize fail cases and limitations of this model.**
>
> As we mentioned in our answer to reviewer arP2, the goal of this paper is to demonstrate the usefulness of learning motion for SSL, which has never been demonstrated yet. Optical flow estimation is simply a pretext task and the goal is not to build a highly specialized flow estimator and produce SOTA flow estimation, but to produce high-quality features with a lot of semantic content. We refer to our answer to reviewer arP2 for a full justification.
>
> **This paper does not show any results of scaling up the model or training data; all backbones reported in table 4 are lightweight. I imagine that training two tasks like this would require a bigger backbone to handle multitasking. A study showing how performance scales with the model size would be useful.**
>
> We refer to our new result with a CNX-B backbone, which shows that the trend we observed at a lower scale is still valid with scaling.
>
> **The authors chose to do training on both tasks on very different datasets. I wonder if training this model on the same video dataset for both tasks would result in features robust enough for video downstream tasks, such as action recognition.**
>
> We agree that not training the motion and content objective on the same source dataset is the main limitation of this paper and we will focus on this goal for future work. The reason we did not tackle this setup first was that learning strong semantic features on video is still a challenge in the video SSL community, while learning strong features from image net is something well established. The goal of this paper was to test the hypothesis whether learning motion can help build better general SSL features, and our current setup and experiments validate this hypothesis which we will further explore in the future by learning from videos only.

---

> > ### Author Response · Authors · 2023-11-20
> > **SOTA comparisons (2/2)**
> >
> > **In section 4.3 (backbone), the authors mention that PWC is not adapted to learn good content features, which explains the low performance in Table 4. However, the method MCRW (in Table 1) uses a PWC backbone and performs much higher than the results of MC-JEPA with a PWC backbone in Table 4. Any reasons behind this significant difference in performance?**
> >
> > We investigated this difference of performance and realized that the number reported in Table 4 corresponds to a model only pretrained on Flow and not in our multi-task setup. The new values for ISeg and VSeg in the multi-task setup are 41.8 and 54.7, which is roughly in the same range as the numbers from MCRW. We will update Table 4.

---

### Official Review · Reviewer_arP2 · 2023-10-28

**Soundness:** 3 good
**Presentation:** 2 fair
**Contribution:** 2 fair
**Rating:** 3
**Confidence:** 4

**Summary:**

This paper proposes to learn a joint embedding for the self-supervision of both motion estimation and image content. They evaluate the proposed model's performance on metrics for optical flow estimation and image/video segmentation.

**Strengths:**

1. The paper is well written.

2. The overall comparison results on two tasks demonstrate the model's superiority especially on segmentation.

**Weaknesses:**

Currently, the provided analysis is not sufficient to demonstrate their contribution by integrating the self-supervised learning for optical flow and content understanding.

1. The proposed method performs on par or slightly worse against other flow estimation methods on Sintel benchmark and Kitti, which can not demonstrate the benefit of the proposed ''Joint-Embedding Predictive Architecture''.  The authors don't give convincing analysis for this issue.

2. Unfair comparison. The proposed method brings more training data compared with the other content methods. It can be seen from Table 2 that the performance in segmentation is likely to degradation as the decrease of motion datasets.

3. Missing analysis. From Table 4, we can see that the model's performance is quite sensitive to the used backbone (more than 10% between Rsenet50 and ConvNext. However, the authors didn't give explanation. Besides, the proposed model uses six loss terms in total for both flow estimation and content learning. I am wondering how to decide the trade-offs and if they would affect the final results.

**Questions:**

See weakness.

---

> ### Author Response · Authors · 2023-11-20
> **Fair comparisons**
>
> Thank you for your comments and useful feedback. We address your concerns below:
>
> **The proposed method performs on par or slightly worse against other flow estimation methods on Sintel benchmark and Kitti, which can not demonstrate the benefit of the proposed ''Joint-Embedding Predictive Architecture''. The authors don't give convincing analysis for this issue.**
>
> Contrary to other flow estimation methods, which have the objective of producing the best quality flow possible, our goal is to leverage flow estimation as a pretext task in order to learn self-supervised representations from images. We therefore choose to keep the flow estimation method simple, and to only make the necessary adjustments that we describe in our answer to reviewer q59r to make it work in a multi-task setup with regular self-supervised learning of semantic content features. We compare the quality of our produced flow to show that it is of comparable quality than popular unsupervised flow estimation methods, without using a complex list of tricks.
>
>
> **Unfair comparison. The proposed method brings more training data compared with the other content methods. It can be seen from Table 2 that the performance in segmentation is likely to degradation as the decrease of motion datasets.**
>
> The other content methods are only trained on ImageNet, which has no motion information. The goal of this paper is to demonstrate the usefulness of learning motion for SSL, and thus bringing optical flow estimation from video datasets, and this is demonstrated by comparing MC-JEPA to VICReg and showing strong improvement on segmentation tasks. The ablation of Table 2 further shows that regardless of the specific flow estimation dataset, a strong improvement is still observed. Indeed for example on Pascal VOC linear frozen segmentation, VICReg performs 60.1 mIoU. MC-JEPA variants with different flow datasets perform between 66.8 and 67.1 which is a very strong improvement, with a very low variance when the flow dataset changes.
>
> **Missing analysis. From Table 4, we can see that the model's performance is quite sensitive to the used backbone (more than 10% between Rsenet50 and ConvNext. However, the authors didn't give explanation. Besides, the proposed model uses six loss terms in total for both flow estimation and content learning. I am wondering how to decide the trade-offs and if they would affect the final results.**
>
> In Table 4 we compare 3 backbones, PWC-Net, ResNet-50 and ConvNeXt-T. For flow estimation all these backbones perform similarly. However, for semantic content learning, PWC-Net, which is a very shallow network specialized for flow estimation, and it therefore does not perform well on segmentation downstream tasks. The gap in performance between ResNet-50 and ConvNeXt-T is not something specific to this paper, and is observed even in supervised learning on ImageNet. ConvNeXt is an architecture that was designed to perform as good as modern vision transformers and it is not surprising to see such gaps for segmentation tasks. We indeed choose ConvNeXt over ResNet from the beginning for this specific reason.
>
> We refer to the explanation we gave to reviewer q59r on how we tune the coefficients of our losses.

---

### Official Review · Reviewer_RRax · 2023-10-31

**Soundness:** 3 good
**Presentation:** 2 fair
**Contribution:** 3 good
**Rating:** 6
**Confidence:** 3

**Summary:**

This paper proposes a self-supervised approach, MC-JEPA, which uses a shared encoder to learn optical flow and content features. The proposed approach achieves good performance on optical flow estimation and images and videos segmentation.

**Strengths:**

This paper presents a valuable and novel insight: self-supervised learning of optical flow estimation and content features can be effectively unified within a single architecture under a multi-task setting. MC-JEPA not only learns motion features from multiple video datasets but also learns content features from large-scale image datasets. This dual-focus approach shows excellent performance across multiple evaluation benchmarks, demonstrating strong generalization capabilities that can be applied to a variety of downstream tasks, from motion prediction to content understanding.

**Weaknesses:**

The writing of the paper needs to further improve. The captions of the figures and tables are too detailed. It is better to condence the captions.

**Questions:**

1. In multi-task learning, how are the weights for different loss functions chosen and adjusted? Will adjusting coefficients for the six different loss functions for each task introduce a significant tuning cost during training? Would it be possible to draw comparisons between this tuning approach and other methodologies in the field of multi-task learning?

2.Although some experiments have already been conducted to demonstrate improvements in Optical Flow Estimation, Image Segmentation, and Video Segmentation, additional experiments could be included to further validate the effectiveness of the proposed methods. For example, more experiments on video object segmentation (YoutubeVOS), video semantic segmentation, video panoptic segmentation(Cityscapes-VPS ,VIPSeg ).

3. What's the motivation of the multi-task learning ? Learning optical flow seems to be a low level visual understanding, why it benefits semantic segmentation?

---

> ### Author Response · Authors · 2023-11-20
> **New segmentation results and motivations**
>
> Thank you for your comments and useful feedback. We address your concerns below:
>
> **The writing of the paper needs to further improve. The captions of the figures and tables are too detailed. It is better to condense the captions.**
>
> The objective was to have self-contained captions, which could be fully understood without diving into the main text. We will change the approach and simplify the captions.
>
> **In multi-task learning, how are the weights for different loss functions chosen and adjusted? Will adjusting coefficients for the six different loss functions for each task introduce a significant tuning cost during training? Would it be possible to draw comparisons between this tuning approach and other methodologies in the field of multi-task learning?**
>
> We refer to the explanation from reviewer q59r on how we tune the coefficients of our losses. Our approach is simple and does not require a lot of tuning.
>
> **2.Although some experiments have already been conducted to demonstrate improvements in Optical Flow Estimation, Image Segmentation, and Video Segmentation, additional experiments could be included to further validate the effectiveness of the proposed methods. For example, more experiments on video object segmentation (YoutubeVOS), video semantic segmentation, video panoptic segmentation(Cityscapes-VPS ,VIPSeg ).**
>
> We decided to focus on linear frozen segmentation (Pascal, Cityscapes, ADE20k) and non-parametric evaluation (DAVIS 2017), which we believe are more suited to evaluate SSL representations. We nevertheless provide new experiments were we fine-tune our model on YoutubeVOS and Cityscapes-VPS:
>
> |			YoutubeVOC (mJ&F)		 | Cityscapes-VPS (VPQ) |
> | -- | -- |
> |VICReg	 |	80.4				               | 58.6 |
> |DINO	 |	83.4				               | 61.1 |
> |MC-JEPA |	84.5				                | 61.9 |
>
> Our results further demonstrate the superiority of the flow estimation multi-task over regular semantic content learning.
>
>
> **What's the motivation of the multi-task learning ? Learning optical flow seems to be a low level visual understanding, why it benefits semantic segmentation?**
>
> Optical flow estimation is indeed a low-level task, and the goal of this paper is to test the following hypothesis: can a low-level task help ground self-supervised training to help learn higher quality high-level representations ? To our surprise, the answer is yes, which we show by comparing MC-JEPA with VICReg and demonstrating strong improvements across all segmentations downstream tasks. Our intuition is that optical flow estimation in this multi-task setup helps structuring the representations with a locality bias, and observing with which pixel is moving where, helps understanding to which object this pixel belongs.

---

> > ### Comment · Reviewer_RRax · 2023-11-22
> >
> > The authors addressed all my concerns. However, the comments from other reviewers pointed out some issues with the experimental results and novelty in the paper, which is worth considering. So I won't raise my score and I decided to keep it unchanged.

---

### Official Review · Reviewer_q59r · 2023-11-01

**Soundness:** 3 good
**Presentation:** 3 good
**Contribution:** 3 good
**Rating:** 5
**Confidence:** 4

**Summary:**

The current focus in self-supervised learning of visual representations has been on capturing content features, which do not include object motion or location information. On the other hand, optical flow estimation is a task that does not require understanding the content of the images. In this work, they introduce MC-JEPA, a joint-embedding predictive architecture, and self-supervised learning approach that combines both objectives to learn the optical flow and content features together. This paper shows that these two objectives benefit from each other, resulting in content features that incorporate motion information. The proposed approach achieves comparable performance with existing unsupervised optical flow benchmarks and common self-supervised learning methods on downstream tasks like semantic segmentation of images and videos.

**Strengths:**

The proposed MC-JEPA method combines self-supervised optical flow estimation and content feature learning in a multi-task setup with a shared encoder. This approach offers several advantages:

1. Joint learning of motion and content features: By integrating optical flow estimation with self-supervised learning, MC-JEPA enables the simultaneous learning of motion information and content features within a single encoder. This allows for the incorporation of motion information into content representations.

2. Improved optical flow estimation: The MC-JEPA method enhances the estimated optical flow by combining it with the self-supervised learning objective. By jointly optimizing these two objectives, the quality of the estimated flow is improved, leading to more accurate motion representations.

3. Transferability to downstream tasks: The content features learned by MC-JEPA transfer well to various downstream tasks, such as optical flow benchmarks and image/video segmentation. This demonstrates the effectiveness of the joint learning approach in producing features that are useful for a wide range of visual tasks.

4. Multi-task learning and joint-embedding architecture: MC-JEPA leverages the benefits of multi-task learning and joint-embedding architectures. By learning multiple tasks simultaneously and using a shared encoder, the method provides a more reliable and generalizable approach to building visual representations.

In summary, MC-JEPA combines the advantages of self-supervised learning, optical flow estimation, multi-task learning, and joint-embedding architecture, resulting in improved motion features and content representations that benefit various visual tasks.

**Weaknesses:**

While the proposed method incorporates some novel self-supervised approaches for image and video learning, the overall architectural novelty is not clearly explained. It is important to clarify the specific innovation of the proposed method in integrating existing self-supervised techniques.

Additionally, as a multitask method, it is crucial to explain how the different tasks are adjusted and why some task coefficients are the same. Providing a clear explanation of the task adjustment strategy and the rationale behind the equal coefficients for certain tasks is necessary.


Regarding the architectural novelty of the proposed method, it is essential to clarify how it integrates existing self-supervised techniques in a unique way. While the specific details of the architecture are not mentioned in the given text, it is important to provide a clear description of how the joint-embedding predictive architecture (MC-JEPA) differs from existing architectures. This could include details about the specific network components, the fusion mechanism for combining optical flow estimation and content feature learning, or any other architectural innovations that distinguish MC-JEPA from previous approaches.

Regarding the multitasking aspect of the method, it is crucial to explain how the different tasks are adjusted and why some task coefficients are the same. This could involve discussing the overall objective function used for multitask learning and how the weights or coefficients for individual tasks are determined. Additionally, providing a rationale for why certain tasks have equal coefficients could be based on their relative importance or the desired balance between different objectives. It is important to clearly explain these aspects to provide a comprehensive understanding of the method.

In summary, to enhance the clarity and completeness of the proposed method, it is necessary to provide a more detailed explanation of the architectural novelty and the rationale behind the task adjustment strategy, including the equal coefficients for certain tasks.

**Questions:**

See weaknesses.

---

> ### Author Response · Authors · 2023-11-20
> **Novelty and balancing loss coefficients**
>
> Thank you for your comments and useful feedback. We address your concerns below:
>
> **1) Novelty.** The main novelty of our paper is more about the setup (multi-task, flow estimation and semantic content learning), and the discovery that flow estimation helps segmentation downstream tasks in SSL, rather than the specific architectural details. However we list here the different new components that we introduced in the paper:
>
> Architecture:
> - Adaptation of the PWC-Net general principle to work with a custom ConvNeXt backbone. We start with the PWC-Net architecture which uses a specific backbone that we replace with a ConvNeXt backbone. This replacement is necessary to make the content learning part work, and requires to replace the first convolutional layers of ConvNeXt by custom layers in order to make the resolution of the feature maps double at each new level. The replacements are described in Appendix C.
>
> - Flow estimator network architecture. We start from the PWC-Net flow estimator and regularize its output with a normalization layer. This is an essential component for the multi-task setup and the flow estimation network is very unstable without it. In addition we increase the number of convolutional filters and number of layers, which shows a positive scaling trend and improves the quality of the produced flow. We present an ablation on network size normalization method in Table 3 and give more details in Appendix C.
>
> Loss functions:
> - New cycle consistency loss. We introduce a new cycle consistency loss that forces f(f^{-1}(x)) = x, where f is the flow and x is the feature map. This loss has a positive impact on the flow estimation quality, which is studied in Figure 5 (2), and is introduced in this paper for the first time.
>
> - New regression and regularization losses. The new multi-task setup we propose was unstable at first and required the addition of flow regression losses at every layer, as well as variance covariance regularization losses. We are the first to make this setup work and this an improvement over regular SSL methods such as VICReg which applies the regularization only at the last layer, and flow estimation methods which apply a reconstruction loss only at the last layer as well.
>
> In addition to these architectural novelties, as we mentioned we focus on the setup and bring various new elements such a study on different multi-task setups in Table 5 and a study on training with different datasets in Table 2.
>
> **2) Loss coefficient tuning.** The MC-JEPA method consists in two tasks: a flow estimation task that learns motion and a SSL general task that learns semantic content. Figure 5 (3) shows how we adjust the coefficient balancing these two tasks. The caption was not clear as it refers to Eq.7 which has no coefficient. We will update Eq.7 with a balancing coefficient $\alpha$ depicted in Figure 5 (3). In our experiments we test several values that are all depicted in the figure, and take the best value, $\alpha=0.1$, which gives the best performance for both flow estimation and segmentation. The flow estimation task is further divided into several loss functions. These loss functions are applied at different levels of the convolutional backbone, and we balance these losses using different sets of coefficients for the different levels. The general idea is to decrease the value of the coefficients as we go deeper in the backbone, in order to focus more on flow reconstruction at the highest resolutions, while still enforcing a coherent flow at the lowest resolutions.
>
> At the first layer, we fix all these coefficients to 1.0, except for the variance loss coefficient which we fix to 25.0 following the VICReg paper. For the deeper layers, we progressively decrease the coefficients of all the flow losses together, and decrease variance and covariance regularization coefficients even further as these are less and less required to maintain stability. In our experiments, we found that the specific values of these coefficients do not matter as long as the general principles we described are followed.

---

### Meta-Review · Area_Chair_C8Ah · 2023-12-05

**Metareview:**

This paper presents a joint-embedding predictive architecture (JEPA) method to learn motion and regular visual features jointly.

After the rebuttal and AC-reviewer discussion stage, the final scores of this paper are 3/5/5/6. The most negative reviewer (rating 3) did not show up in the discussion, while the other three reviewers confirmed to maintain their original rating. Meanwhile, the only positive reviewer (rating 6) said that he would not defend the acceptance of this paper. Overall, the average score of this paper falls below the acceptance threshold, and the AC found no reason to overturn the recommendation.

**Justification For Why Not Higher Score:**

The average score falls below the acceptance threshold.

**Justification For Why Not Lower Score:**

N/A

---

### Decision · Program_Chairs · 2024-01-16

Reject